# Consumption of Unprocessed and Ultraprocessed Foods in Adolescents with Obesity: Associations with Neuroendocrine Mediators of Appetite Regulation and Binge Eating Symptoms

**DOI:** 10.3390/nu17233711

**Published:** 2025-11-26

**Authors:** Patrícia Sousa Neres, Aline de Piano Ganen, Raquel Munhoz da Silveira Campos, Joana Pereira de Carvalho Ferreira, Lila Missae Oyama, Ana Raimunda Dâmaso, Deborah Cristina Landi Masquio

**Affiliations:** 1Programa de Mestrado Profissional em Nutrição: do Nascimento à Adolescência, Curso de Nutrição, Centro Universitário São Camilo (CUSC), São Paulo 05025-010, SP, Brazil; patriciasousaneres@gmail.com (P.S.N.); mestradonutricao@saocamilo-sp.br (A.d.P.G.); 2Programa de Pós-Graduação Interdisciplinar em Ciências da Saúde, Departamento de Biociências, Universidade Federal de São Paulo (UNIFESP), Campus Baixada Santista, Santos 11010-150, SP, Brazil; raquel.munhoz@unifesp.br; 3Laboratório Multidisciplinar em Alimentos e Saúde, Faculdade de Ciências Aplicadas, Universidade Estadual de Campinas (UNICAMP), Limeira 13484-350, SP, Brazil; joanacf@unicamp.br; 4Programa de Pós-Graduação em Nutrição, Departamento de Fisiologia, Universidade Federal de São Paulo (UNIFESP), Campus São Paulo, São Paulo 04023-061, SP, Brazil; lmoyama@unifesp.br (L.M.O.); ana.damaso@unifesp.br (A.R.D.)

**Keywords:** obesity, unprocessed food, ultraprocessed food, neuroendocrine mediators, energy balance, adolescent

## Abstract

**Background/Objectives**: Obesity is a multifactorial disease associated with increased consumption of ultraprocessed foods and reduced intake of unprocessed foods. Binge eating, one of the most prevalent eating disorders among adolescents, is closely linked to obesity. Food intake is regulated by both the hedonic system, responsible for reward responses, and the physiological system, which controls hunger and satiety through hormones, such as ghrelin and leptin. The present study aimed to investigate associations between the intake of unprocessed and ultraprocessed foods, neuroendocrine mediators of appetite regulation, and binge eating in adolescents with obesity. **Methods**: This cross-sectional study included 96 adolescents with obesity who were recruited in São Paulo, Brazil, between 2010 and 2012. Anthropometric and body composition assessments were performed. Binge eating symptoms were evaluated using the binge eating scale (BES), and dietary intake was assessed with a validated Food Frequency Questionnaire, with items classified according to the Nova system. Frequency data were converted into annual consumption scores. Serum levels of ghrelin, leptin, neuropeptide Y (NPY), agouti-related peptide (AgRP), melanin-concentrating hormone (MCH), and alpha-melanocyte-stimulating hormone (α-MSH) were analyzed. **Results**: Lower consumption of unprocessed foods was associated with higher ghrelin concentrations (*p* = 0.023), accompanied by a greater percentage of body fat (*p* = 0.047) and a reduced percentage of lean mass (*p* = 0.047) compared with adolescents in the second tertile. AgRP was a positive predictor of annual consumption score of ultraprocessed food (β = 0.30; *p* = 0.04), independent of age, body fat, and binge eating symptoms. **Conclusions**: In conclusion, lower intake of unprocessed foods was associated with alterations in orexigenic and anorexigenic mediators, suggesting that dietary patterns in adolescents with obesity may influence the neuroendocrine mediators of appetite regulation.

## 1. Introduction

Obesity is a chronic disease characterized by an abnormal or excessive accumulation of body fat. It is a multifactorial condition that involves biological, social, cultural, behavioral, and political factors, as well as ones related to public health. In children and adolescents, the prevalence of obesity has risen dramatically in recent decades, with more than 160 million individuals aged 5–19 years currently affected worldwide [1,2].

Although complex and multifaceted, obesity is closely related to increased consumption of unhealthy foods and sedentary lifestyle [2]. In recent decades, substantial changes in dietary patterns have been observed, particularly the replacement of unprocessed or minimally processed foods with ultraprocessed foods (UPFs). UPFs are energy-dense products rich in sugars, fats, and additives, which are highly palatable and widely consumed during adolescence. The high availability and regular consumption of these products are consistently associated with poor diet quality, which is related to an increased risk of non-communicable diseases, including obesity [3,4].

Adolescence is a critical developmental period marked by increased autonomy in food choices, heightened sensitivity to reward, and a tendency toward unhealthy eating behaviors [5]. During this phase, inadequate intake of unprocessed and minimally processed foods, especially fruits and vegetables, contributes to reduced fiber and micronutrient intake. Conversely, there is a higher consumption of UPFs during this phase, raising concerns about the health implications of this dietary pattern in health and disease [6,7]. Such dietary patterns may not only affect energy balance but also probably interfere with neuroendocrine regulation of appetite [8,9].

The regulation of appetite and food consumption is controlled by the interplay between homeostatic mechanisms, which integrate metabolic signals of hunger and satiety, and hedonic pathways, which drive the motivation to consume palatable foods. Orexigenic neuropeptides, such as neuropeptide Y (NPY) and agouti-related peptide (AgRP), as well as melanin-concentrating hormone (MCH), promote food intake and weight gain, whereas anorexigenic peptides, such as α-melanocyte stimulating hormone (α-MSH), exert an anorexigenic effect. Peripheral hormones, including ghrelin and leptin, modulate these hypothalamic pathways, linking dietary exposure to central appetite regulation [10,11]. In the context of obesity, sustained hyperleptinemia results in central leptin resistance, which blunts the hormone’s inhibitory effects on orexigenic neurons and its stimulatory effects on anorexigenic pathways, ultimately leading to dysregulation of appetite control and increased food intake, as demonstrated in Figure 1 [10,11]. In environments characterized by easy access to UPFs, hedonic signals may override homeostatic control, reinforcing excessive intake and contributing to obesity development [5,12].

While binge eating symptoms are common among adolescents with obesity, the relationship between dietary patterns, eating behaviors, and neuroendocrine mediators remains unclear [13,14]. The study conducted by Ayton et al. (2021) explored the consumption of ultraprocessed foods and binge eating in the general population, but not in adolescents [15].

Most available studies have focused either on UPF consumption and metabolic alterations [4,16,17] or on mental health outcomes [18], but few have integrated the level of food processing with the neuroendocrine control of appetite in adolescents. Little is known about how consumption of foods according to processing level relates to orexigenic and anorexigenic mediators, such as AgRP, NPY, MCH, and α-MSH in this age group. The hypothesis of the present study is that a higher frequency of ultraprocessed foods and lower frequency of unprocessed and minimally processed food in the diet is associated with alterations in neuroendocrine mediators involved in hunger and satiety regulation in adolescents with obesity. Therefore, the present study aimed to investigate the associations between annual consumption scores of unprocessed/minimally processed and ultraprocessed foods, neuroendocrine mediators of appetite regulation, and binge eating symptoms in adolescents with obesity.

## 2. Materials and Methods

This was a descriptive analytical cross-sectional study involving adolescents participating in the Obesity Study Group at the Federal University of São Paulo (UNIFESP), as part of a 1-year interdisciplinary obesity treatment program. Data collection took place between 2010 and 2012. Recruitment was performed through public outreach campaigns using traditional media channels (television, radio, and newspapers) in São Paulo, Brazil.

Adolescents were eligible to participate if they were between 14 and 19 years old, had obesity, defined as a BMI above the 95th percentile for age and sex according to CDC reference curves [19], and were in the post-pubertal stage (Tanner Stage V) [20]. A clinical evaluation conducted by an endocrinologist confirmed inclusion criteria and screened for exclusion criteria. Exclusion criteria included being in the pubertal transition phase, having endocrine or genetic disorders, pregnancy, chronic alcohol consumption, previous medication use, or musculoskeletal or clinical limitations preventing participation in the interdisciplinary program.

A total of 122 adolescents were initially recruited; however, after verification of complete baseline data, 96 participants were retained for analysis in the present study. Exclusions were due to incomplete data records. This project was approved by the Research Ethics Committee (COEP) of UNIFESP (number 5,862,487). All participants and/or guardians were informed about the procedures involved in data collection, as well as the risks and benefits of all procedures. After agreeing to participate, all parents/participants signed the Informed Consent Form (ICF) and/or the Assent Form (AF).

Anthropometric evaluation was conducted through standardized measurements of body weight (kg), height (m), and waist circumference, with body mass index (BMI, kg/m^2^) subsequently calculated in accordance with established procedures [21]. Body weight was measured on a Filizola scale with an accuracy of 0.1 kg (PL 180 model, Filizola S/A, São Paulo, SP, Brazil), and height was measured against a wall-mounted stadiometer, with a measurement scale of 0.5 cm (Sanny, model ES 2030, São Bernardo do Campo, SP, Brazil). BMI was calculated by dividing body weight (kg) by height (m) squared. BMI percentiles and z-scores were calculated according to age- and sex-specific reference standards [19]. Waist circumference was measured at the midpoint between the last rib and the iliac crest [21]. Body composition was estimated by air displacement plethysmography using the BOD POD system (version 1.69; Life Measurement Instruments) [22].

Visceral and subcutaneous fat were estimated by abdominal ultrasound using a multifrequency 3.5 MHz transducer (broadband). Subcutaneous fat was defined as the distance between the skin and the superficial plane of the rectus abdominis muscle. Visceral fat was defined as the distance between the deep plane of the same muscle and the anterior wall of the aorta [23].

Neuroendocrine mediators of appetite regulation were assessed by blood collection in a clinical laboratory after an overnight fast of approximately 8 h, via peripheral vein puncture in the forearm. Blood samples were frozen at −80 °C and analyzed later. Serum levels of total ghrelin, leptin, neuropeptide Y (NPY), agouti-related peptide (AgRP), melanin-concentrating hormone (MCH), and alpha-melanocyte-stimulating hormone (α-MSH) were measured using RD Systems enzyme-linked immunosorbent assay kits (Minneapolis, MN, USA). The orexigenic NPY/AgRP ratio was calculated. Reference values for leptin were considered according to Gutin et al. [24], with hyperleptinemia identified when leptin concentrations were >20 ng/mL in boys and >24 ng/mL in girls.

Binge eating symptoms were assessed using the validated binge eating scale (BES) questionnaire. It consists of 16 items, each containing 3 to 4 response options, with respondents choosing the one that best represents them [25]. Scores on this questionnaire range from 0 to 46. Individuals with a score ≤ 17 were classified as not presenting binge eating symptoms, whereas those with scores ≥18 were classified as presenting binge eating symptoms, with further distinction into moderate symptoms (18–26) and severe symptoms (≥27) [25].

Regarding dietary intake assessment, food consumption was evaluated using a Food Frequency Questionnaire (FFQ) [26]. The FFQ used in this study has been previously applied and tested in Brazilian adolescents and has demonstrated validity, reproducibility, and calibration [27,28]. The instrument recorded the frequency of consumption of 94 food items over the previous six months and was self-administered, with parental assistance provided when necessary to minimize recall bias. To ensure standardized completion and improve data quality, the questionnaire was filled out in the presence of a trained research nutritionist, who provided clarifications on portion sizes, food items, and response categories when needed. In addition, the researcher verified that all items were completed and that no response fields were left blank before the questionnaire was accepted. Frequency was assessed in seven categories: (1) never; (2) less than once a month; (3) 1–3 times a month; (4) once a week; (5) 2–4 times a week; (6) once a day; and (7) 2 or more times a day [26]. The reporting of this observational study followed the STROBE-nut (Strengthening the Reporting of Observational Studies in Epidemiology—Nutrition) guidelines, which provide specific recommendations for dietary assessment in nutritional epidemiology [29].

FFQ food items were grouped according to processing level based on the Nova classification, which distinguishes foods according to the extent and purpose of industrial processing [3,30]. Unprocessed and minimally processed foods are obtained directly from plants or animals and subjected to minimal alterations, such as cleaning, grinding, drying, fermentation, pasteurization, refrigeration, freezing, or vacuum packaging, without the addition of substances like salt, sugar, oils, or fats [3,30]. Ultraprocessed foods (UPFs) are industrial formulations typically made with little or no whole foods, composed mainly of substances extracted from foods (oils, fats, sugar, starch, proteins), derived from food constituents (hydrogenated fats, modified starch), or synthesized in laboratories (flavorings, colorings, emulsifiers, thickeners, artificial sweeteners) [3,30]. In this study, the FFQ items classified in these categories are described in Table 1. For foods for which a clear definition of the category could not be found, the criteria described by Martinez-Steele et al. [30] were used.

The annual consumption score (ACS) was calculated according to methodology proposed by Fornes et al. [31]. Each frequency category from the FFQ was converted into a conversion (weighting) factor representing the number of consumption events per year, according to the following Formula (1):ACS = (1/365) × [(a + b)/2](1)
where “a” and “b” correspond to the number of days each reported frequency of consumption represents over the course of one year. For example, when the response category was “1 to 3 times per month”, “a” was set at 12 (reflecting one occurrence per month across 12 months), while “b” was set at 36 (reflecting three monthly occurrences across a year). For frequencies of 1 or more times per day, the score was standardized as 1. The conversion (weighting) factors used to calculate the ACS are presented in Table 2.

In the present study, ACS was calculated for unprocessed and minimally processed foods, as well as for ultraprocessed foods, by summing the score of all food items in each group. It should be noted that this scoring method does not account for portion sizes, focusing exclusively on the frequency of food consumption. However, this method is considered a valuable tool to investigate dietary patterns and their associations with health outcomes, supporting the relevance of applying consumption scores in epidemiological research [32]. Missing data from FFQ was handled by excluding questionnaires with more than 20% unanswered items, while isolated missing responses were treated as non-consumption [31,32].

Statistical analysis was performed using JAMOVI^®^ software, version 2.3.28, considering *p* < 0.05 as significant. The Shapiro–Wilk normality test was used to assess the distribution of variables. One-way ANOVA and Kruskal–Wallis tests were applied to compare tertile groups according to the annual consumption score of ultraprocessed food (Tertile 1 ≤ 0.48; Tertile 2 > 0.48 ≤ 6.68; Tertile 3 > 6.68) and unprocessed/minimally processed food (Tertile 1 ≤ 5.55; Tertile 2 >5.55 ≤ 8.39; Tertile 3 > 8.39).

Simple linear regression models were performed to explore the associations between food consumption scores (ultraprocessed and unprocessed/minimally processed foods, treated as continuous variables) and the neuroendocrine mediators of appetite regulation (AgRP, NPY, adiponectin, leptin, ghrelin, MCH, and α-MSH). Each biomarker was entered as an independent variable in separate models. Multiple linear regression models were performed to investigate the association between continuous scores of ultraprocessed and unprocessed/minimally processed food consumption and the independent variables of interest. Two models were created, in which the dependent variables (outcomes) were (1) the annual consumption score of ultraprocessed foods and (2) the annual consumption score of unprocessed/minimally processed foods. The independent variables included were age (years), body fat percentage or visceral fat (cm), neuroendocrine mediators of appetite regulation (AGRP, NPY, ghrelin, MCH, and α-MSH), and binge eating symptoms (BES, yes/no) were entered as covariates. All models met the required assumptions for multiple regression analysis. Multicollinearity was not detected, as variance inflation factor (VIF) values were <10 and tolerance was >0.80. Independence of residuals was confirmed by Durbin–Watson statistics within the acceptable range (1.5–2.5). Normality, linearity, and homoscedasticity of residuals were verified through visual inspection of standardized plots (QQ plots). No influential cases were identified, as Cook’s distance values were <1 for all observations.

The statistical power of the study was calculated according to the sample used (*n* = 96). Post hoc power analyses were conducted using G*Power (v.3.1.9.7), with α = 0.05 and the observed effect sizes from the analyses performed. For the one-way ANOVA comparing tertiles, the achieved power was 1 − β = 0.56. The multiple linear regression models, which included four predictors and an observed effect size of $f^{2}=0.25$, demonstrated a statistical power of 1 − β = 0.98.

## 3. Results

### 3.1. Binge Eating Symptoms

The sample consisted of 96 adolescents with obesity, including 54 girls and 42 boys. According to the dichotomized BES classification, 64.6% (*n* = 62) of the participants did not present binge eating symptoms, whereas 35.4% (*n* = 34) exhibited such symptoms. Among those with binge eating, 7.3% (*n* = 7) were classified as having severe symptoms, and 28.1% (*n* = 27) as having moderate symptoms (Figure 2).

### 3.2. Comparison of Adolescents According to Tertiles of Unprocessed and Ultraprocessed Food

Table 3 presents the age, annual food consumption scores according to processing level, binge eating symptoms, and anthropometric profile of participants, stratified by tertiles of ultraprocessed and unprocessed/minimally processed food consumption. When adolescents were categorized according to tertiles of annual ultraprocessed food consumption, significant differences were observed in the consumption scores across processing levels. Processed food (H = 19.2, *p* < 0.001) and ingredient scores (H = 17.7, *p* < 0.001) differed significantly across tertiles. No significant differences were found for BES scores or anthropometric variables (*p* > 0.05). Regarding tertiles according to the annual unprocessed/minimally processed food consumption score, the median score of ultraprocessed food increased across tertiles (H = 10.9, *p* = 0.004). A similar pattern was observed for processed food (H = 22.8, *p* < 0.001) and ingredient scores (H = 20.0, *p* < 0.001), both of which rose significantly with higher consumption tertiles. Moreover, participants in tertile 1 showed a significantly higher percentage of body fat (F = 3.23, *p* = 0.046) and a lower percentage of lean mass (F = 3.22, *p* = 0.047) compared with those in the tertile 2.

Table 4 presents the values of neuroendocrine mediators of appetite regulation according to tertiles of annual consumption score of ultraprocessed and unprocessed/minimally food. No significant differences were observed between tertiles of ultraprocessed food. However, when participants were analyzed according to the consumption scores of unprocessed/minimally processed foods, participants in the lowest tertile (tertile 1) showed higher ghrelin concentrations compared to tertile 2 (H = 6.57; *p* = 0.038).

### 3.3. Predictors of Ultraprocessed and Unprocessed/Minimally Consumption Score

The results of the simple linear regression models are shown in Table 5. For ultraprocessed food consumption, none of the neuroendocrine mediators reached statistical significance, although AgRP demonstrated a trend toward a positive association (β = 0.262; *p* = 0.079). Similarly, no significant associations were observed between unprocessed/minimally processed food consumption and the biomarkers analyzed.

Linear multiple regression analyses were conducted to identify predictors of ultraprocessed and unprocessed/minimally processed food consumption scores (Table 6). For the ultraprocessed food consumption score, the final model, including age, body fat percentage, AgRP, and BES score explained **5**% of the total variance (R^2^ = 0.05). AgRP was a significant positive predictor (β = 0.30; 95% CI: 0.001–0.61; *p* = 0.04), independent of another variables. Each one-unit increase in AGRP was associated with a 0.30-point increase in the ultraprocessed food score (95% CI: 0.001–0.61; *p* = 0.04) (Table 6). For the unprocessed/minimally processed food consumption score, no significant associations were observed for any of the predictors analyzed. Overall, the results indicate no statistically significant associations of neuroendocrine mediators of appetite regulation (NPY, ghrelin, MCH, α-MSH, and leptin), adiposity measures (body fat and visceral fat), and eating behavior (BES) with food consumption scores.

## 4. Discussion

In the present study, binge eating symptoms were identified in approximately one-third of adolescents with obesity. Although eating disorders and obesity are strongly related with poor diet quality in adolescents [33,34], we did not observe direct associations of binge eating symptoms with UPF consumption and neuroendocrine mediators of appetite regulation in adolescents with obesity.

We observed higher ghrelin concentrations among adolescents with lower consumption of unprocessed/minimally processed foods (tertile 1) compared to tertile 2. However, no significant differences were found between tertile 1 and tertile 3, suggesting that the association may not follow a strictly linear dose–response pattern. This result may reflect a threshold effect, in which reductions in natural food intake below a certain level do not lead to further increases in ghrelin. Alternatively, the effect size may be modest, and the present sample may not provide sufficient power to detect small between-group differences. Despite these limitations, the overall trend suggests that lower consumption of fiber- and micronutrient-rich natural foods may be associated with impaired satiety signaling and increased orexigenic drive [35].

This aligns with findings from De Ruyter et al. [36], who reported inverse associations between ghrelin, diet quality, and psychosocial well-being in children and adolescents, reinforcing the hypothesis that ghrelin is responsive not only to metabolic cues, but also to the qualitative aspects of diet.

Ghrelin is an orexigenic hormone, released by enteroendocrine cells of the stomach, that increases during fasting and decreases during feeding, exerting an orexigenic effect on appetite regulation [35]. Ghrelin acts beyond homeostatic appetite control, influencing reward pathways, stress responses, and food motivation through receptors located outside the hypothalamus.

By enhancing the drive to consume highly palatable foods, ghrelin may promote preference for ultraprocessed products regardless of energy needs [35]. Overall, low intake of unprocessed foods may intensify orexigenic signaling, whereas diets rich in minimally processed foods, typically higher in fiber and micronutrients, appear to favor anorexigenic responses, support gut–brain regulation, and reduce hedonic overeating triggered by ultraprocessed foods [37,38].

Our regression models further demonstrated that the orexigenic neuropeptide AgRP was a positive predictor of annual consumption score of ultraprocessed food, independent of adiposity, age, and binge eating symptoms. This reinforces the hypothesis that orexigenic mediators may influence food choices, increasing preference for ultraprocessed products. Although body fat percentage showed only a borderline association, the direction of this relationship suggests a possible reinforcing cycle between adiposity, hormonal regulation, and unhealthy eating patterns [39].

Both simple and multiple regression models yielded predominantly non-significant associations, indicating that the frequency of consumption of ultraprocessed and unprocessed/minimally processed foods was not linearly related to others appetite-regulating hormones. This suggests that the interaction between food processing level and neuroendocrine regulation of appetite may probably involve additional metabolic, behavioral, or dietary mediators that were not accounted for in the present analysis [37,38].

In the literature, UPFs have been proposed to influence brain circuits involved in appetite regulation, particularly orexigenic pathways. Their high palatability, energy density, and concentration of refined sugars, fats, and food additives can stimulate mesolimbic dopaminergic reward circuits, leading to an enhanced hedonic response to food [40,41]. This mechanism may be amplified during adolescence, a period characterized by reward hypersensitivity and immature inhibitory control [42]. Moreover, chronic exposure to UPFs alters the gut–brain axis through microbiota disruption, increased intestinal permeability, and low-grade systemic inflammation. These alterations impair the regulation of appetite hormones, potentially intensifying orexigenic signaling and reinforcing maladaptive eating patterns [40,41,42]. Such mechanisms may explain the observed association between AgRP and UPF intake in our regression models, independent of adiposity. However, these findings should be interpreted with caution, as no other mediators of appetite regulation showed significant associations with the frequency of UPF consumption.

Dietary patterns low in unprocessed foods were also associated with unfavorable body composition. Adolescents in the lowest tertile of unprocessed/minimally processed food intake exhibited a significantly higher percentage of body fat and a lower percentage of lean mass compared with those in the second tertile. This suggests that a dietary pattern poor in natural foods may contribute to unfavorable body composition. These findings reinforce the view that diet quality exerts an influence beyond energy intake, affecting the balance between fat and lean tissue, which is critical in the prevention of metabolic complications during adolescence [33,42].

It is noteworthy that adolescents in the highest tertile of UPF consumption also showed higher scores of unprocessed/minimally processed food intake. Rather than reflecting dietary substitution, this pattern likely indicates a generally higher overall food intake. Therefore, the associations observed should be interpreted within the context of global eating behavior rather than as mutually exclusive consumption patterns [43,44].

Previous studies conducted in samples of adolescents with obesity from the same research group have already documented a high prevalence of metabolic comorbidities, particularly insulin resistance (58.0%), hyperleptinemia (66.7%), and metabolic syndrome (27.8%) [45]. These conditions may influence appetite regulation and could partially explain inter-individual variability in neuroendocrine responses [11,46]. However, the present study did not include comorbidity analyses, and future investigations are needed to explore whether these metabolic alterations modify or mediate the associations observed in mediators of appetite control.

The findings observed in the present study suggest that increasing the intake of unprocessed and minimally processed foods could represent a relevant nutritional approach to modulate orexigenic signaling and support appetite regulation in adolescents with obesity. Interventions focusing solely on reducing UPFs may be insufficient if not accompanied by strategies that simultaneously promote whole food intake, satiety, and neuroendocrine control of appetite [43,44].

We did not identify human studies published to date that investigated the association of dietary patterns characterized by foods according to processing level with alterations in orexigenic and anorexigenic neuropeptides and binge eating. This gap limits our ability to fully elucidate the biological pathways by which diet quality influences appetite regulation and energy balance, underscoring the need for translational research that integrates nutritional epidemiology with neuroendocrine biomarkers.

One limitation of the present study was the use of a Food Frequency Questionnaire (FFQ), which, although appropriate for assessing long-term dietary patterns, does not allow precise quantification of caloric or nutrient intake. Because the instrument relies on self-reported memory and estimated frequency of consumption, recall and reporting bias cannot be fully excluded, particularly in adolescents. However, to minimize this limitation, the FFQ was completed in the presence of parents and trained nutritionists, who provided standardized guidance and clarified portion sizes when necessary. Even so, the dietary data should be interpreted as an indicator of habitual food consumption rather than exact energy or macronutrient intake.

The cross-sectional design further does not permit to infer causality, limiting conclusions to associations rather than directionality of effects. In addition, the restricted clinical sample and modest sample size may limit external generalizability and reduce statistical power to detect subtle associations, particularly in subgroup analyses. Post hoc power analyses confirmed that the one-way ANOVA across tertiles demonstrated limited statistical power (1 − β = 0.56), reflecting small between-group effect sizes. However, the multiple linear regression models showed adequate statistical power (1 − β = 0.98), supporting the robustness of the association findings. Finally, the assessment of appetite-related neuroendocrine mediators was based on single fasting measurements, which do not capture their dynamic postprandial fluctuations.

## 5. Conclusions

In this clinical sample of adolescents with obesity, lower consumption of unprocessed and minimally processed foods was associated with higher fasting ghrelin concentrations, suggesting greater orexigenic activity, while orexigenic neuropeptide AgRP remained a positive predictor of ultraprocessed food intake. Additionally, dietary patterns with low frequency of unprocessed and minimally processed foods may contribute to higher body fat and lower lean mass. Although exploratory in nature, the results reinforce the relevance of promoting diets based on unprocessed foods during adolescence, not only to improve overall diet quality but also to influence biological pathways involved in appetite control and obesity management.

## Figures and Tables

**Figure 1 nutrients-17-03711-f001:**
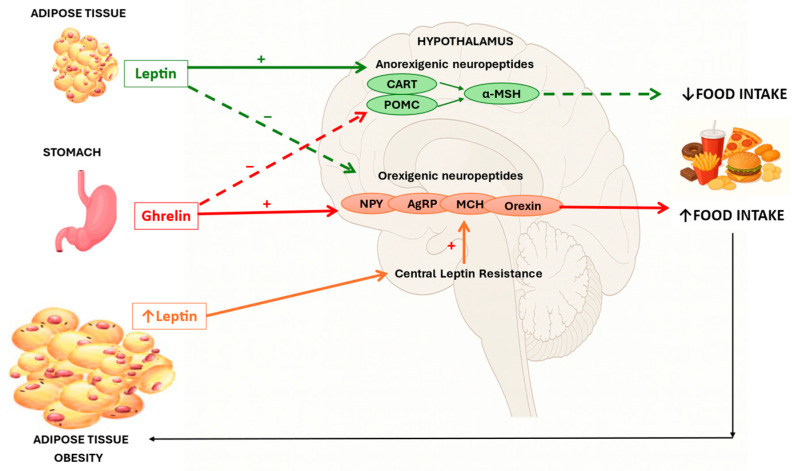
Neuroendocrine pathways linking peripheral signals of energy balance (leptin and ghrelin) to hypothalamic regulation of appetite and food intake in obesity. The green arrow represents the action of leptin on anorexigenic and orexigenic neuropeptide expression. The red arrow represents the action of ghrelin on these pathways. The orange arrow indicates central leptin resistance, reducing activation of the anorexigenic pathway. A solid line indicates stimulation (+), whereas a dashed line indicates inhibition (−). CART: cocaine- and amphetamine-regulated transcript; POMC: pro-opiomelanocortin; α-MSH: alpha-melanocyte-stimulating hormone; MCH: melanin-concentrating hormone; NPY: neuropeptide Y; AgRP: agouti-related peptide. Adapted from: Dâmaso et al., 2024 [11].

**Figure 2 nutrients-17-03711-f002:**
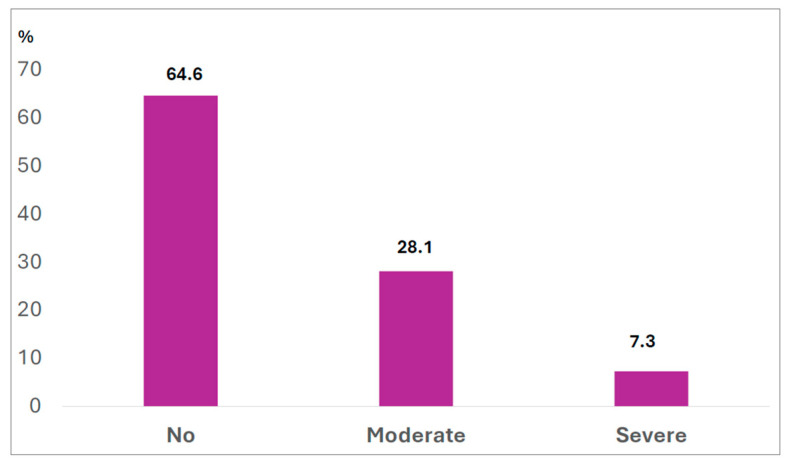
Classification of binge eating symptoms in adolescents with obesity.

**Table 1 nutrients-17-03711-t001:** Classification of foods into unprocessed/minimally and ultraprocessed categories according to the NOVA system [3,30].

NOVA Classification	Items from Food Frequency Questionary
Unprocessed food	Whole milk, skimmed milk, natural yogurt, lettuce, kale/cabbage, watercress/arugula, cauliflower, beetroot, spinach/collard greens, peas, tomato, carrot, coffee, green corn, potato, boiled cassava, orange/tangerine, banana, pineapple, apple/pear, papaya, strawberry, avocado, melon/watermelon, grape, mango, cooked rice, cooked beans, chicken, beef, fish, pork, coffee, mate tea (chimarrão)
Ultraprocessed and minimally processed food	Potato chips or savory snacks, chocolate/brigadeiro, plain or packaged cake, ice cream (tub or popsicle), powdered chocolate drink, candies, cheeseburger (beef or chicken), cheese bread, hot dog, diet yogurt, cream cheese, mayonnaise, margarine, plain biscuits, filled biscuits, breakfast cereals, processed meats, sausage, frankfurter, soft drink (regular), diet soft drink, flavored mate tea, artificial juices, sweetener, mousse-type desserts, chocolate croissant, ham and cheese croissant, fermented milk drink

**Table 2 nutrients-17-03711-t002:** Conversion factors derived from the frequency categories of the Food Frequency Questionnaire (FFQ) used to compute the annual consumption score (ACS), according to Fornes et al. (2002) [31].

Consumption Frequency	Conversion Factor Used to Calculate Annual Consumption Score (0–1)
Never	0.00
Less than once a month	0.02
1–3 times per month	0.07
Once per week	0.14
2–4 times per week	0.43
Once per day	1.00
Two or more times per day	1.00

**Table 3 nutrients-17-03711-t003:** Anthropometric profile, binge eating symptoms, and annual consumption scores of food intake in adolescents with obesity according to tertiles of ultraprocessed and unprocessed/minimally processed food.

Ultraprocessed Food Consumption Score				
	Tertile 1 (*n* = 33) Median 2.91 (Range 0.48–4.25)	Tertile 2 (*n* = 31) Median 5.34 Range (4.29–6.68)	Tertile 3 (*n* = 32) Median 8.20 Range (6.71–17.6)	*p* ^1^
Age (years)	16.51 ± 1.85	17.25 ± 1.97	16.90 ± 1.46	0.349
BES score ^2^	16 (1–46)	12 (3–32)	16.5 (0–30)	0.376
**Annual consumption score**				
Unprocessed/minimally processed food score	6.06 (1.35–14.9) b	6.40 (1.66–14.7)	8.17 (4.00–23.4)	**0.006**
Ingredients score	0.45 (0.07–3.00) b	1.07 (0.02–3.00) c	1.65 (0.02–3.00)	**<0.001**
Processed food score	1.79 (0.23–4.08) a,b	2.50 (0.64–5.32) c	3.03 (1.27–9.59)	**<0.001**
**Anthropometry**				
Body weight (kg)	101.00 (71.60–162.90)	98.40 (77.50–155.80)	96.70 (76.80–145.70)	0.867
Height (m)	1.68 ± 0.10	1.69 ± 0.09	1.68 ± 0.09	0.857
BMI (kg/m^2^)	34.50 (28.30–48.50)	34.80 (29.40–45.50)	34.40 (28.20–48.10)	0.969
BMI percentile	98.8 (96.5–99.7)	98.7 (95.9–99.9)	98.7 (96.0–99.9)	0.453
BMI z score	2.27 ± 0.27	2.19 ± 0.33	2.23 ± 0.32	0.603
Body fat (%)	42.93 ± 5.95	44.38 ± 6.08	44.13 ± 6.34	0.594
Lean mass (%)	57.07 ± 5.95	55.62 ± 6.09	55.87 ± 6.34	0.594
Body fat (kg)	44.25 ± 11.63	44.86 ± 9.67	44.55 ± 12.16	0.975
Lean mass (kg)	58.20 ± 10.95	56.16 ± 11.11	55.37 ± 8.77	0.517
Visceral fat (cm)	4.38 ± 1.11	4.35 ± 1.45	4.68 ± 1.28	0.785
Subcutaneous fat (cm)	4.12 ± 1.05	4.23 ± 3.99	3.99 ± 0.73	0.535
Waist circumference (cm)	99.83 ± 10.86	98.57 ± 10.34	98.05 ± 9.51	0.785
**Unprocessed/Minimally Food Consumption Score**				
	**Tertile 1 (*n* = 32)** **Median 3.94** **Range (1.35–5.55)**	**Tertile 2 (*n* = 32)** **Median 6.80** **Range (5.65–8.39)**	**Tertile 3 (*n* = 32)** **Median** **Range 10.9 (8.46–23.4)**	***p*** ^1^
Age (years)	16.80 ± 2.17	16.78 ± 1.40	17.04 ± 1.72	0.646
BES score ^2^	13 (0–33)	15 (3–43)	15 (2–37)	0.908
**Annual consumption score**				
Ultraprocessed food score	4.93 (0.48–8.24) b	4.38 (1.15–12.4)	6.87 (0.62–17.6)	**0.004**
Ingredient score	0.28 (0.02–2.39) b	1.13 (0.07–3.00) c	1.89 (0.02–3.00)	**<0.001**
Processed food score	1.62 (0.33–3.51) b	2.25 (0.23–6.01) c	3.49 (1.27–6.01)	**<0.001**
**Anthropometry**				
Body weight (kg)	101.00 (71.60–162.90)	98.40 (77.50–155.80)	96.70 (76.90–145.70)	0.967
Height (m)	1.67 ± 0.09	1.70 ± 0.09	1.67 ± 0.09	0.262
BMI (kg/m^2^)	34.50 (28.30–48.50)	34.80 (29.40–45.50)	34.40 (28.20–48.10)	0.307
BMI percentile	98.8 (95.9–99.9)	98.7 (96.0–99.9)	98.4 (95.9–99.8)	0.608
BMI z score	2.28 ± 0.32	2.22 ± 0.29	2.20 ± 0.31	0.591
Body fat (%)	45.22 ± 5.02a	41.55 ± 6.67	44.63 ± 6.01	**0.046**
Lean mass (%)	54.78 ± 5.02a	58.45 ± 6.67	55.38 ± 6.01	**0.047**
Body fat (kg)	45.40 ± 9.57	42.40 ± 11.23	45.84 ± 12.31	0.422
Lean mass (kg)	54.71 ± 9.71	59.03 ± 10.49	56.07 ± 10.49	0.234
Visceral fat (cm)	4.79 ± 1.46	4.22 ± 0.98	4.39 ± 1.31	0.199
Subcutaneous fat (cm)	4.26 ± 0.87	3.84 ± 0.93	4.24 ± 0.88	0.125
Waist circumference (cm)	101.45 ± 10.20	96.91 ± 9.85	98.12 ± 10.19	0.193

Parametric variables are described as mean ± standard deviation. Non-parametric variables as median (minimum and maximum). ^1^ One-way ANOVA for parametric variables and Kruskal–Wallis test for non-parametric variables; a represents significant difference between tertile 1 × tertile 2; b represents significant difference between tertile 1 × tertile 3; and c represents significant difference between tertile 2 × tertile 3 Values in bold indicate statistically significant differences (*p* < 0.05). ^2^ BES, binge eating scale; BMI, body mass index. Key finding: Body fat (%) was significantly higher in tertile 1 of unprocessed/minimally processed food consumption score compared to tertile 2. Lean mass (%) was significantly lower in tertile 1 of unprocessed/minimally processed food consumption score compared to tertile 2.

**Table 4 nutrients-17-03711-t004:** Neuroendocrine mediators of appetite regulation in adolescents with obesity according to tertiles of ultraprocessed and unprocessed/minimally processed food.

Ultraprocessed Food Consumption Score				
	Tertile 1 (*n* = 33)	Tertile 2 (*n* = 31)	Tertile 3 (*n* = 32)	*p* ^1^
AgRP (ng/mL)	0.34 (0.12–5.24)	0.96 (0.13–2.71)	0.82 (0.12–14.50)	0.434
NPY (ng/mL)	1.23 (0.50–30.90)	2.10 (0.48–8.12)	2.08 (0.54–12.90)	0.249
NPY/AGRP ratio	3.17 (0.47–11.70)	2.54 (10.70–6.99)	2.80(0.41–11.68)	0.674
Leptin (ng/mL)	35.25 (7.78–61.74)	33.45 (49.56–124.38)	34.31 (85.61–80.26)	0.945
Ghrelin (ng/mL)	1.17 (0.20–1.53)	1.05 (0.88–1.44)	1.00 (0.31–1.44)	0.221
MCH (ng/mL)	4.77 (1.68–10.84)	6.40 (1.45–1.93)	5.93 (1.36–21.10)	0.638
α-MSH (ng/mL)	0.76 (0.07–3.63)	1.84 (0.17–9.14)	1.63 (0.25–6.44)	0.428
**Unprocessed/Minimally Food Consumption Score**				
	**Tertile 1 (** ***n* = 32)**	**Tertile 2 (** ***n* = 32)**	**Tertile 3 (** ***n* = 32)**	***p*** ^1^
AgRP (ng/mL)	0.45 (0.12–3.23)	0.67 (0.12–10.66)	0.38 (0.13–14.48)	0.911
NPY (ng/mL)	1.56 (0.55–8.12)	1.69 (0.60–9.61)	1.54 (0.48–30.89)	0.916
NPY/AGRP ratio	2.86 (0.47–11.68)	1.94 (0.41–11.70)	3.11 (0.67–9.83)	0.165
Leptin (ng/mL)	34.24 (1.67–61.19)	34.11 (4.96–62.59)	35.24 (14.75–124.38)	0.344
Ghrelin (ng/mL)	1.18 (0.31–1.44) a	0.96 (0.20–1.49)	1.11 (0.53–1.44)	**0.038**
MCH (ng/mL)	7.21 (1.36–10.24)	4.88 (1.50–11.60)	5.50 (1.88–21.10)	0.713
α-MSH (ng/mL)	0.85 (0.30–6.44)	1.53 (0.24–3.31)	1.16 (0.07–9.14)	0.955

Parametric variables are described as mean ± standard deviation. Non-parametric variables as median (minimum and maximum); AgRP, agouti-related peptide; α-MSH, alpha-melanocyte-stimulating hormone; MCH, melanin-concentrating hormone; NPY, neuropeptide Y; Parametric variables are described as mean ± standard deviation. Non-parametric variables as median (minimum and maximum). ^1^ One-way ANOVA for parametric variables and Kruskal–Wallis test for non-parametric variables; a represents significant difference between tertile 1 × tertile 2; Value in bold indicates statistically significant difference (*p* < 0.05). Key finding: Ghrelin concentration was significantly higher in tertile 1 of unprocessed/minimally processed food consumption score compared to tertile 2.

**Table 5 nutrients-17-03711-t005:** Simple regression analysis for determinants of ultraprocessed and unprocessed/minimally processed food consumption in adolescents with obesity.

Ultraprocessed Food Consumption Score							
				95% Confidence Interval			
	Estimate	R^2^	Standard Error	Lower Limit	Upper Limit	t	*p*
AgRP	0.262	0.03	0.147	−0.03	0.555	1.78	0.079
NPY	0.011	1.79	0.087	−0.161	0.184	0.130	0.897
Adiponectin	0.026	0.02	0.029	−0.015	0.067	1.240	0.217
Leptin	6.48	1.66	0.018	−0.036	0.038	0.034	0.973
Ghrelin	−2.10	0.05	1.40	−4.920	0.711	−1.510	0.139
MCH	0.069	0.001	0.111	−0.151	0.291	0.632	0.530
α-MSH	0.342	0.026	0.217	−0.088	0.772	1.580	0.118
**Unprocessed/Minimally Processed Food Consumption Score**							
				**95% Confidence** **Interval**			
	**Estimate**	**R^2^**	**Standard Error**	**Lower Limit**	**Upper Limit**	**t**	** *p* **
AgRP	0.123	0.000	0.183	−0.240	0.486	0.675	0.502
NPY	0.166	0.025	0.105	−0.043	0.376	1.580	0.118
Adiponectin	0.024	0.000	0.025	−0.027	0.075	0.920	0.360
Leptin	0.023	0.014	0.023	−0.022	0.069	1.02	0.313
Ghrelin	−1.200	0.010	1.750	−4.730	2.32	−0.687	0.496
MCH	0.053	0.002	0.137	−0.221	0.326	0.382	0.703
α-MSH	0.224	0.001	0.268	−0.308	0.757	0.836	0.405

Note: AgRP, agouti-related peptide; α-MSH, alpha-melanocyte-stimulating hormone; MCH, melanin-concentrating hormone; NPY, neuropeptide Y. Key finding: No significant association was observed between neuroendocrine mediators of appetite regulation and ultraprocessed and unprocessed/minimally processed food consumption score.

**Table 6 nutrients-17-03711-t006:** Multiple regression analysis for determinants of ultraprocessed and unprocessed/minimally processed food consumption in adolescents with obesity.

Ultraprocessed Food Consumption Score							Unprocessed/Minimally Processed Food Consumption Score						
			95% Confidence Interval							95% Confidence Interval			
Predictors	Estimate	Standard Error	Lower Limit	Upper Limit	t	*p*	Predictors	Estimate	Standard Error	Lower Limit	Upper Limit	t	*p*
R^2^ = 0.05							R^2^ = 0.01						
**Intercept**	0.84	2.18	0.024	8.68	1.99	0.04	**Intercept**	7.58	4.55	16.63	16.63	1.66	0.10
**Age**	0.19	0.18	−0.17	0.56	1.05	0.29	**Age**	0.09	0.23	−0.36	0.55	0.41	0.68
**Body fat (%)**	0.02	0.05	−0.08	0.13	0.51	0.61	**Body fat (%)**	−0.04	0.06	−0.18	0.08	−0.71	0.47
**AgRP**	0.30	0.15	0.001	0.61	1.99	**0.04**	**AgRP**	0.10	0.19	−0.27	0.48	0.56	0.57
**BES (Yes-No)**	0.29	0.67	−1.03	1.63	0.44	0.66	**BES (Yes-No)**	0.53	0.83	−1.12	2.20	0.64	0.52
**R^2^ = 0.07**							**R^2^ = 0.03**						
**Intercept**	2.89	3.61	−4.23	10.07	0.80	0.42	**Intercept**	7.28	4.37	−1.40	15.96	1.66	0.09
**Age**	0.14	0.18	−0.22	0.52	0.79	0.43	**Age**	0.12	0.22	−0.32	0.57	0.55	0.58
**Body fat (%)**	0.01	0.05	−0.10	0.11	0.12	0.89	**Body fat (%)**	−0.05	0.06	−0.18	0.07	−0.89	0.37
**NPY**	0.01	0.08	−0.15	0.19	0.20	0.83	**NPY**	0.16	0.10	−0.04	0.38	1.54	0.12
**BES (Yes-No)**	0.30	0.68	−1.05	1.66	0.44	0.65	**BES (Yes-No)**	0.37	0.82	−1.26	2.02	0.45	0.64
**R^2^ = 0.11**							**R^2^ = 0.07**						
**Intercept**	−2.72	4.48	−11.77	6.33	−0.60	0.54	**Intercept**	5.98	5.73	−5.59	17.55	1.04	0.30
**Age**	0.03	0.19	−0.35	0.42	0.18	0.85	**Age**	0.24	0.24	−0.25	0.74	0.99	0.32
**Body fat (%)**	0.16	0.08	−0.01	0.33	1.98	0.05	**Body fat (%)**	−0.04	0.10	−0.26	0.17	−0.99	0.66
**Ghrelin**	−0.26	0.50	−1.27	0.75	−0.52	0.60	**Ghrelin**	0.25	0.64	−2.10	0.48	−1.26	0.21
**BES (Yes-No)**	0.67	0.84	−1.02	2.36	0.79	0.43	**BES (Yes-No)**	−0.81	1.07	−1.92	2.42	0.23	0.81
**R^2^ = 0.01**							**R^2^ = 0.01**						
**Intercept**	3.04	4.56	−6.07	12.15	0.66	0.50	**Intercept**	11.42	5.63	0.17	22.67	2.02	0.04
**Age**	0.10	0.23	−0.36	0.57	0.44	0.65	**Age**	−0.06	0.28	−0.63	0.51	−0.21	0.83
**Body fat (%)**	0.01	0.06	−0.10	0.14	0.30	0.76	**Body fat (%)**	−0.07	0.07	−0.22	0.08	−0.90	0.37
**MCH**	0.08	0.11	−0.14	0.31	0.74	0.45	**MCH**	0.04	0.14	−0.24	0.32	0.29	0.77
**BES (Yes-No)**	0.29	0.85	−1.41	2.01	0.34	0.73	**BES (Yes-No)**	0.58	1.05	−1.53	2.69	0.55	0.58
**R^2^ = 0.04**							**R^2^ = 0.01**						
**Intercept**	1.11	3.64	−6.13	8.36	0.30	0.76	**Intercept**	7.63	4.53	−1.36	16.64	1.68	0.09
**Age**	0.17	0.18	−0.19	0.53	0.93	0.35	**Age**	0.09	0.22	−0.36	0.54	0.39	0.69
**Body fat (%)**	0.02	0.05	−0.08	0.13	0.46	0.64	**Body fat (%)**	−0.05	0.06	−0.18	0.08	−0.76	0.44
α-**MSH**	0.39	0.22	−0.04	0.84	1.76	0.59	α-**MSH**	0.52	0.83	−1.13	0.75	0.72	0.46
**BES (Yes-No)**	0.36	0.66	−0.96	1.69	0.54	0.08	**BES (Yes-No)**	0.20	0.27	−0.35	2.17	0.63	0.53
**R^2^ = 0.04**							**R^2^ = 0.01**						
**Intercept**	1.87	4.38	−6.87	10.63	0.42	0.67	**Intercept**	7.02	5.45	−3.86	17.97	1.28	0.20
**Age**	0.31	0.26	−0.20	0.83	1.19	0.23	**Age**	−0.01	0.32	−0.65	0.64	−0.02	0.98
**Visceral fat**	−0.27	−0.27	0.29	0.31	−0.93	0.35	**Visceral fat**	−0.11	0.36	−0.85	0.62	−0.31	0.75
**Leptin**	−0.01	0.01	0.01	0.03	−0.28	0.77	**Leptin**	0.02	−0.02	−0.02	0.94	0.94	0.34
**BES (Yes-No)**	0.79	0.81	−0.82	2.42	0.98	0.32	**BES (Yes-No)**	0.20	1.01	−1.81	2.22	0.20	0.84

AgRP, agouti-related peptide; α-MSH, alpha-melanocyte-stimulating hormone; MCH, melanin-concentrating hormone; NPY, neuropeptide Y; BES, binge eating symptoms; Value in bold indicates statistically significant difference (*p* < 0.05). Key finding: Ultraprocessed food consumption score was significantly associated with AgRP, independent of age, body fat (%), and binge eating symptoms.

## Data Availability

The data presented in this study are available on request from the corresponding author. The data are not publicly available due to privacy and ethical restrictions related to the protection of participant information.

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
