# Peer review of "Consumption of Unprocessed and Ultraprocessed Foods in Adolescents with Obesity: Associations with Neuroendocrine Mediators of Appetite Regulation and Binge Eating Symptoms"

_nutrients, 2025, doi:10.3390/nu17233711_

Round 1
Reviewer 1 Report
Comments and Suggestions for Authors
The effect on eating patterns is a potential important mechanism for the overall negative health impact of UPF. Adolescence is a critical time for the development of a healthy body composition and for healthy eating patterns. The study is therefore addressing a critical research question.
The reader would benefit from a clearer description of the neurobiological mechanisms controlling hunger and satiety, for example by adding a graphic with the organs and compounds involved and the direction of the effects.
The methodology of the study should be better explained. The study hypothesis should be clearly formulated and the sample size required to prove it with adequate power calculated. The paper does not clearly state whether the hypothesis is that a diet containing a higher proportion of UPF is leading to alterations of neurobiological mechanisms and then to overconsumption and altered body composition or instead higher consumption is the result of an altered balance of the homeostatic mechanisms.
I would like to challenge the term used of “biomarker of energy balance” used to describe neurobiological mediators of appetite control. With the former, the reader thinks of markers of energy intake and energy expenditure.
The population should be described : what is the distribution of BMIs in the whole group ? Table 2 does not explain the significance of the different values, but one could guess that m±sd are provided in some cases and median and ranges (?) in other cases. If the latter is correct, than the BMI is included between 28.3 and 48.5 kg/m2, which indicates that some children were mildly obese and some were severely obese. BMI percentile are more appropriate than BMI per se to describe the level of obesity of the children. Neurobiological mediators of appetite control may differ according to the level of adiposity. Additionally, the reader would expect comments on the presence of obesity comorbidities (including hypertension, dyslipidaemia, hyperglycaemia, non-alcoholic fatty liver disease, and metabolic syndrome).
I found the calculation of the Annual Consumption Score obscure. The paper used to convert FFQ categories into score (reference 31) is in Portuguese, but I understand that the calculation does not actually provide a score, but a conversion factor (a weight) used to calculate the score for each individual. The heading of the second column of Table 1 should be changed accordingly e.g. into “conversion factor” or “weighting factor” used to calculate ACS, as the ACS is the sum of all the scores obtained.
Please also note that PubMed has different Authors for reference 31:
Fornes NS, Martins IS, Velasquez-Melendez G, Latorre Mdo R. Escores de consumo alimentar e níveis lipêmicos em população de São Paulo, Brasil [Food consumption scores and serum lipids levels in the population of São Paulo, Brazil]. Rev Saude Publica. 2002 Feb;36(1):12-8. Portuguese. doi: 10.1590/s0034-89102002000100003. PMID: 11887224.
Please use “tercile” and “tertile” consistently.
Use a table to list the foods included among the UPF and the processed, rather than listing them in the text.
I find the analysis inconsistent. The definition of the model has the consumption of UPF or MPF as outcome variables (consistent with a hypothesis of mediators leading to food preferences) but then the analysis is done with tertiles, suggesting that the tertile of consumption is the independent variable.
The statement “For UPF the median score increased progressively from the lowest to the highest tertile”, with a p level calculated, is redundant, as it implicit in the division into tertiles. In fact, the median and range of scores should be indicated in the headlines of the columns.
The individuals in the highest tertile of UPF are also having a high level of consumption of MPF. This should be explained, and conditions the whole interpretation of the data.
Please be consistent in the definition of the categories of foods. I believe the second part of Table 2 refers to unprocessed and minimally processed foods and not to unprocessed foods only.
Table 3 mentions binge eating symptoms in the title, but the values are not provided in the table. The p value for AgrP is not provided, and this is the basis of the concluding remarks.
The finding of a significantly different value of ghrelin between the low tertile of unprocessed/minimally processed food consumption and the middle tertile is used as an argument to support the relationship between the two variables. Why then the highest tertile has non significant differences with the lowest tertile ?
I do not support the analysis in Table 4. I think an analysis of the relationship between the variables, e.g. though a simple regression, should have been done.
Furthermore, the analysis is difficult to interpret, as there is no biological rationale of e.g. a mid tertile being significantly different from the first and the third tertile, while the first and the third are not significantly different.
For the multiple regression lines, what would the final models look like and what would be the total variance explained ? In addition, rather than highlighting that AGRP came out as a significant predictor of UPF consumption, in combination with body fat, age and the presence of BES (again, how much variance explained?) I would comment on the absolute lack of significance of all the other measures. This observation is the basis of the main study result, and I do not think the data warrant such conclusion.
Similarly, another conclusion is made on the basis of the AgrP findings. In addition of the p value not provided in Table 3, an explanation should be given as to why tertile 1 has a lower level than tertiles 2 and 3, and there is no significant difference between tertiles 2 and 3. Is there a threshold effect or is it a data issue ?
The arguments used in the discussion are contradictory and not warranted by the overall profile of the data. The statement that “A relevant finding of our study was the higher ghrelin concentrations in adolescents with lower consumption of unprocessed and minimally processed foods” (Line 324) is based on results reported in the second part of Table 3 (for unprocessed/minimally processed foods) : tertile 1 (1,20 ng/mL); tertile 2 (0,96 ng/mL); tertile 3 (1,11 ng/mL). One would expect that tertile 3 would also be lower than tertile 1, but it is not. I would look for potential outliers in the tertile 1 data distribution.
This statement is followed by another statement that “Elevated ghrelin among those with lower intake of UPF suggests that a diet poor in natural sources of fiber and micronutrient my impair satiety regulation, increasing hunger signaling and preference for more palatable foods (Line 331). However, in Table 3 (first part, on UPF) the level of ghrelin in individuals with lower intake of UPF is not significantly different from that of the other two tertiles.
In the view of the Authors, which is the group with highest ghrelin concentration – this with low intake of unprocessed/minimally processed foods or those with low levels of UPF ?
The comments on the regression models should really focus on the negative results, rather than just the isolated finding of AgRP.
The statement that “UPF can exert a strong influence on brain circuits regulating appetite, particularly orexogenic pathways” (Line 392) is not justified by the findings and by the way the analysis of the data was designed.
Author Response
REVIEWER 1
Comments: The effect on eating patterns is a potential important mechanism for the overall negative health impact of UPF. Adolescence is a critical time for the development of a healthy body composition and for healthy eating patterns. The study is therefore addressing a critical research question.
The reader would benefit from a clearer description of the neurobiological mechanisms controlling hunger and satiety, for example by adding a graphic with the organs and compounds involved and the direction of the effects.
Answer: Thank you very much for this valuable suggestion. In the revised version, we have included a schematic representation illustrating the main neurobiological pathways regulating hunger and satiety. The figure 1 summarizes the interaction between peripheral organs (stomach and adipose) and central nervous system structures (mainly hypothalamic nuclei), as well as the direction of hormonal and neuropeptidergic signaling involved in appetite regulation.
The methodology of the study should be better explained. The study hypothesis should be clearly formulated and the sample size required to prove it with adequate power calculated. The paper does not clearly state whether the hypothesis is that a diet containing a higher proportion of UPF is leading to alterations of neurobiological mechanisms and then to overconsumption and altered body composition or instead higher consumption is the result of an altered balance of the homeostatic mechanisms.
Answer: We thank the reviewer for this important and constructive observation. We agree that the formulation of the study hypothesis and the methodological description required further clarification. To address this, we have now explicitly stated the study hypothesis in the Introduction: The hypothesis of the present study is that a higher frequency of ultra-processed foods and lower frequency of unprocessed food in the diet is associated with alterations in neuroendocrine mediators involved in hunger and satiety regulation in adolescents with obesity.
Statistical power was calculated and has now been included in the Methods section, under Statistical Analysis.
I would like to challenge the term used of “biomarker of energy balance” used to describe neurobiological mediators of appetite control. With the former, the reader thinks of markers of energy intake and energy expenditure.
Answer: Thank you for this insightful comment. We agree that the term “biomarkers of energy balance” may be misleading, as it is typically associated with markers of energy intake and expenditure. To improve conceptual clarity and accurately reflect the physiological role of the variables examined, we have replaced this expression throughout the manuscript with the term “neuroendocrine mediators of appetite regulation.”
The population should be described : what is the distribution of BMIs in the whole group ? Table 2 does not explain the significance of the different values, but one could guess that m±sd are provided in some cases and median and ranges (?) in other cases. If the latter is correct, than the BMI is included between 28.3 and 48.5 kg/m2, which indicates that some children were mildly obese and some were severely obese. BMI percentile are more appropriate than BMI per se to describe the level of obesity of the children. Neurobiological mediators of appetite control may differ according to the level of adiposity. Additionally, the reader would expect comments on the presence of obesity comorbidities (including hypertension, dyslipidaemia, hyperglycaemia, non-alcoholic fatty liver disease, and metabolic syndrome).
Answer: We appreciate the reviewer’s observation. We have added a note in the footnote of Table 3 clarifying that values for parametric variables are expressed as mean ± standard deviation, while non-parametric variables are expressed as median (minimum–maximum). In addition, we have included BMI percentile and z-score data in the sample description.
Data on cardiometabolic comorbidities (e.g., hypertension, dyslipidaemia, altered glucose metabolism, NAFLD, and metabolic syndrome) were collected but were not included in the present manuscript because they are being analyzed in a separate study with a different primary outcome. However, we agree that these conditions are highly prevalent among adolescents with obesity, and this point has now been included in the Discussion section. We have also cited previously published studies from our research group reporting the prevalence of these comorbidities in similar samples.
I found the calculation of the Annual Consumption Score obscure. The paper used to convert FFQ categories into score (reference 31) is in Portuguese, but I understand that the calculation does not actually provide a score, but a conversion factor (a weight) used to calculate the score for each individual. The heading of the second column of Table 1 should be changed accordingly e.g. into “conversion factor” or “weighting factor” used to calculate ACS, as the ACS is the sum of all the scores obtained.
Answer: We appreciate your observation. In the revised version, we have clarified in the Methods section that the values presented in Table 1 correspond to conversion (weighting) factors derived from the reference study (Ref. 31), which were used to calculate the Annual Consumption Score (ACS) for each participant. The heading of the second column in Table 1 has been modified accordingly to “Conversion factor used to calculate ACS”, as well as, the title of Table 1.
Please also note that PubMed has different Authors for reference 31:
Answer: Thank you very much for the observation. The list of authors for Reference 31 has been verified and corrected according to the official PubMed record.
- Fornes NS, Martins IS, Velasquez-Melendez G, Latorre Mdo R. Escores de consumo alimentar e níveis lipêmicos em população de São Paulo, Brasil [Food consumption scores and serum lipids levels in the population of São Paulo, Brazil]. Rev Saude Publica. 2002 Feb;36(1):12-8. doi: 10.1590/s0034-89102002000100003
Please use “tercile” and “tertile” consistently.
Answer: We thank the reviewer for this observation. The terminology has been revised throughout the manuscript to use the term “tertile” consistently.
Use a table to list the foods included among the UPF and the processed, rather than listing them in the text.
Answer: We thank the reviewer for this helpful suggestion. In the revised version, we have included a new table (Table 2) summarizing the foods classified as ultra-processed and unprocessed/minimally processed food according to the NOVA classification, instead of listing them in the text.
I find the analysis inconsistent. The definition of the model has the consumption of UPF or MPF as outcome variables (consistent with a hypothesis of mediators leading to food preferences) but then the analysis is done with tertiles, suggesting that the tertile of consumption is the independent variable.
Answer: We thank the reviewer for this observation. We apologize for any lack of clarity in the description of the analytical strategy. In the descriptive analysis (Tables 3 and 4), variables were compared across tertiles of UPF and unprocessed/minimally processed food consumption. However, in the linear regression models (Tables 5 and 6), the continuous ACS score (for UPF and unprocessed foods, respectively) was used as the dependent variable, while age, body fat percentage, appetite-related neuropeptides, and binge eating symptoms were included as independent predictors. The text has been revised to make this distinction clearer.
The statement “For UPF the median score increased progressively from the lowest to the highest tertile”, with a p level calculated, is redundant, as it implicit in the division into tertiles. In fact, the median and range of scores should be indicated in the headlines of the columns.
Answer: We agree that the statement describing the progressive increase across tertiles is redundant, as this pattern is inherent to the tertile classification. Accordingly, we removed this sentence from the text. In addition, we have included the median and range (minimum–maximum) values of the UPF and unprocessed food consumption scores in the column headers of the respective tables.
The individuals in the highest tertile of UPF are also having a high level of consumption of MPF. This should be explained, and conditions the whole interpretation of the data.
Answer: The overlap between high consumption of ultraprocessed foods and high consumption of unprocessed/minimally processed foods suggests that some adolescents present a generally higher total intake rather than mutually exclusive dietary patterns. This pattern is consistent with previous findings in adolescent populations, where higher UPF intake frequently coexists with higher overall energy intake, including foods from other processing levels. We have now clarified this point in the Discussion section, highlighting that the associations observed should not be interpreted as substitution effects, but rather as potential markers of broader eating behavior patterns (e.g., hyperphagia, higher frequency of meals/snacks, or greater caloric exposure).
Please be consistent in the definition of the categories of foods. I believe the second part of Table 2 refers to unprocessed and minimally processed foods and not to unprocessed foods only.
Answer: We agree that the second part of Table 2 refers to both unprocessed and minimally processed foods, according to the NOVA classification. The table title and related text have been revised accordingly to ensure consistency in terminology throughout the manuscript.
Table 3 mentions binge eating symptoms in the title, but the values are not provided in the table. The p value for AgrP is not provided, and this is the basis of the concluding remarks.
Answer: We thank you for this careful observation. The p-value for AgRP has now been added to Table 3 (now named Table 4). In addition, we adjusted the title of the table to ensure consistency between the title and the data presented.
The finding of a significantly different value of ghrelin between the low tertile of unprocessed/minimally processed food consumption and the middle tertile is used as an argument to support the relationship between the two variables. Why then the highest tertile has non significant differences with the lowest tertile ?
Answer: Thank you for this thoughtful comment. We acknowledge that the difference in ghrelin levels observed between the lowest and middle tertiles of unprocessed/minimally processed food consumption was not consistent across all tertiles. We agree that this isolated finding does not provide a clear biological gradient or rationale between variables. Therefore, this interpretation was revised in the revised version of the manuscript to avoid overinterpretation of a non-linear or inconsistent pattern.
I do not support the analysis in Table 4. I think an analysis of the relationship between the variables, e.g. though a simple regression, should have been done. Furthermore, the analysis is difficult to interpret, as there is no biological rationale of e.g. a mid tertile being significantly different from the first and the third tertile, while the first and the third are not significantly different.
Answer: We thank the reviewer for this valuable comment. We acknowledge the limitations of the analysis presented in Table 4 and we decided to exclude it from our manuscript. Following this recommendation, we performed additional simple linear regression analyses to examine the relationships between food consumption scores (dependent variable) and the neuroendocrine profile of appetite regulation. These results are now presented in the new Table 5.
For the multiple regression lines, what would the final models look like and what would be the total variance explained ? In addition, rather than highlighting that AGRP came out as a significant predictor of UPF consumption, in combination with body fat, age and the presence of BES (again, how much variance explained?) I would comment on the absolute lack of significance of all the other measures. This observation is the basis of the main study result, and I do not think the data warrant such conclusion.
Answer: We thank the reviewer for this important and constructive comment. The final multiple regression models have now been clearly described in the Results section, including the standardized coefficients and the total variance explained (R²) for each model, which are presented in Table 6 and highlighted in red. In the revised Results section, we also emphasize that the only statistically significant model, explained 5% of the variance in the outcome. For unprocessed/minimally processed food consumption, no model reached statistical significance.
Similarly, another conclusion is made on the basis of the AgrP findings. In addition of the p value not provided in Table 3, an explanation should be given as to why tertile 1 has a lower level than tertiles 2 and 3, and there is no significant difference between tertiles 2 and 3. Is there a threshold effect or is it a data issue ?
Answer:
We thank the reviewer for this insightful comment. The p-value for this comparison has now been added to Table 3 (now is Table 4). The AgRP data were non-parametric, and thus the comparisons between tertiles were analyzed using the Kruskal-Wallis test (p=0.434). This analysis did not reveal statistically significant differences between tertiles, although tertile 1 showed lower levels than the other tertiles (with no significance statistical). The absence of significant differences between tertiles 2 and 3 may reflect a threshold effect, in which AgRP levels above a certain point are not associated with further changes in the outcome. Alternatively, it may indicate that the effect size is small and the current sample does not provide sufficient power to detect differences. This explanation has now been added to the revised Discussion section.
The arguments used in the discussion are contradictory and not warranted by the overall profile of the data. The statement that “A relevant finding of our study was the higher ghrelin concentrations in adolescents with lower consumption of unprocessed and minimally processed foods” (Line 324) is based on results reported in the second part of Table 3 (for unprocessed/minimally processed foods) : tertile 1 (1,20 ng/mL); tertile 2 (0,96 ng/mL); tertile 3 (1,11 ng/mL). One would expect that tertile 3 would also be lower than tertile 1, but it is not. I would look for potential outliers in the tertile 1 data distribution.
Answer: We appreciate the reviewer’s careful observation. Following this suggestion, we re-examined the distribution of ghrelin concentrations in tertile 1 of unprocessed/minimally processed food consumption and identified one extreme value that could potentially influence the median. After removing this outlier, the descriptive data were updated in Table 3 (Now Table 4). Importantly, the statistical significance of the differences across tertiles, assessed using the Kruskal–Wallis test, remained unchanged, with significant differences persisting between tertiles 1 and 2. This adjustment and clarification have been incorporated into the revised version of the manuscript and the updated Table 3 (Now Table 4).
This statement is followed by another statement that “Elevated ghrelin among those with lower intake of UPF suggests that a diet poor in natural sources of fiber and micronutrient my impair satiety regulation, increasing hunger signaling and preference for more palatable foods (Line 331). However, in Table 3 (first part, on UPF) the level of ghrelin in individuals with lower intake of UPF is not significantly different from that of the other two tertiles.
Answer: We thank the reviewer for pointing this out. The sentence stating that “Elevated ghrelin among those with lower intake of UPF suggests that a diet poor in natural sources of fiber and micronutrients may impair satiety regulation…” was adjusted to reflect the non-significant differences observed across tertiles. We now emphasize that the observed pattern is a tentative interpretation and should be interpreted with caution.
In the view of the Authors, which is the group with highest ghrelin concentration – this with low intake of unprocessed/minimally processed foods or those with low levels of UPF ?
Answer: In our sample, the highest ghrelin concentrations were observed among adolescents with lower intake of unprocessed/minimally processed foods, rather than among those with low levels of UPF consumption. Although both analyses were based on tertile stratification, the association was more evident when the sample was categorized according to the unprocessed/minimally processed food score. This pattern remained after the exclusion of one outlier and did not change the statistical significance of the Kruskal–Wallis test.
The comments on the regression models should really focus on the negative results, rather than just the isolated finding of AgRP.
Answer: We agree that the interpretation of the regression models should not emphasize only the isolated association observed for AgRP. In the revised version, we have adjusted the Results and Discussion sections to clarify that, overall, the regression analyses did not demonstrate significant linear associations between food processing scores and most of the neuroendocrine or metabolic markers.
The statement that “UPF can exert a strong influence on brain circuits regulating appetite, particularly orexogenic pathways” (Line 392) is not justified by the findings and by the way the analysis of the data was designed.
We agree that the original statement may have implied a causal or mechanistic conclusion that is not directly supported by our study design or findings. The sentence has now been revised and changed, in order to avoid overinterpretation and to better reflect the correlational nature of the results.
Reviewer 2 Report
Comments and Suggestions for Authors
This work addresses an important, timely, and innovative topic that is likely to attract significant interest from both the scientific community and practitioners involved in child and adolescent health.
Point 1 – Introduction
The introduction clearly demonstrates that obesity among youth is a growing phenomenon with serious health consequences. Numerous references to previous research show familiarity with the topic and strengthen credibility. The text moves seamlessly from general information about obesity, through the role of UPF, to the neuroendocrine regulation of appetite and binge eating. In the last part, the authors emphasized the innovative nature of the research undertaken.The literature is correct
Point 2 – Methodology
The study was approved by the committee. It included adolescents from the Obesity Study Group at the Federal University of São Paulo (UNIFESP). The study’s methodology and design are well-planned, robust, and reliable, incorporating both biological and behavioral measures and providing a broad perspective. Its greatest strength lies in the combination of food classification, neuroendocrine biomarkers, and binge eating symptoms—an uncommon and valuable approach. However, the FFQ captures only food frequency, which may underestimate or overestimate actual energy and macronutrient intake. The sample size is relatively small, and despite training, adolescents may have struggled to accurately estimate food frequency. Moreover, the study included only adolescents with obesity participating in the UNIFESP program, which may limit the generalizability of the findings (e.g., to the broader population of adolescents with obesity).The section is described in detail, but the section's introduction lacks information about the study's date. Did all participants complete the study? There are inclusion/exclusion criteria, but there is no data on how many people were initially excluded from the project. Were all questionnaires completed correctly? How was the questionnaire submitted for analysis?
Point 3 – Results
The results are logically structured and clearly organized. Tables are extensive and detailed, allowing readers to analyze the data independently. However, each table would benefit from a brief accompanying statement that highlights the key finding. In the "Binge eating symptoms" section, it is worth adding a graphical presentation of the results.
Point 4 – Discussion
The discussion is substantively strong and well-argued. Its weaknesses include length and repetitiveness; it's worth focusing on the most important elements. Practical implications could be further emphasized – for example, whether the results suggest that nutritional interventions should focus on increasing the consumption of unprocessed foods.
Point 5 – Conclusions Overall, the conclusions are supported by the results but do not fully reflect all findings. It should be clearly stated that the results apply only to this specific group. A stronger link between the conclusions and the methodological limitations would enhance their credibility.
Point 6- The tables are factually correct and complete, a graphical presentation of the results would improve the quality of the work. The literature is correct, consisting mainly of newer reports. It is worth supplementing the abstract with the date of the research.
Author Response
Reviewer 2
This work addresses an important, timely, and innovative topic that is likely to attract significant interest from both the scientific community and practitioners involved in child and adolescent health.
Point 1 – Introduction
The introduction clearly demonstrates that obesity among youth is a growing phenomenon with serious health consequences. Numerous references to previous research show familiarity with the topic and strengthen credibility. The text moves seamlessly from general information about obesity, through the role of UPF, to the neuroendocrine regulation of appetite and binge eating. In the last part, the authors emphasized the innovative nature of the research undertaken.The literature is correct.
Answer: We thank the reviewer for the positive assessment of the Introduction.
Point 2 – Methodology
The study was approved by the committee. It included adolescents from the Obesity Study Group at the Federal University of São Paulo (UNIFESP). The study’s methodology and design are well-planned, robust, and reliable, incorporating both biological and behavioral measures and providing a broad perspective. Its greatest strength lies in the combination of food classification, neuroendocrine biomarkers, and binge eating symptoms—an uncommon and valuable approach. However, the FFQ captures only food frequency, which may underestimate or overestimate actual energy and macronutrient intake. The sample size is relatively small, and despite training, adolescents may have struggled to accurately estimate food frequency.
Answer: We agree that the FFQ provides frequency-based estimates rather than precise quantitative intake values, which may introduce under- or over-reporting. This limitation has now been explicitly addressed in the Discussion. We have also clarified that the FFQ was intentionally selected because it enables the assessment of long-term dietary patterns, which is consistent with the study’s objective of characterizing habitual food consumption according to processing level, rather than estimating exact energy or macronutrient intake. In addition, to minimize reporting bias, the FFQ was completed in the presence of a trained research nutritionist, with parental assistance when necessary, ensuring standardized guidance during data collection.
Moreover, the study included only adolescents with obesity participating in the UNIFESP program, which may limit the generalizability of the findings (e.g., to the broader population of adolescents with obesity). The section is described in detail, but the section's introduction lacks information about the study's date. Did all participants complete the study? There are inclusion/exclusion criteria, but there is no data on how many people were initially excluded from the project. Were all questionnaires completed correctly? How was the questionnaire submitted for analysis?
Answer: We thank the reviewer for these important observations. We acknowledge that the sample consisted exclusively of adolescents with obesity enrolled in a structured clinical program at UNIFESP, which may restrict external generalizability. This point has now been explicitly stated in the Limitations section. We included in the Methods section the period during which data collection took place; the total number of eligible participants initially screened and the number excluded based on eligibility criteria; confirmation that all FFQs were completed in the presence of a trained nutritionist, with parental support when necessary, ensuring completeness and consistency of all responses. Additionally, The FFQ data were checked for completeness before being entered into the database.
Point 3 – Results
The results are logically structured and clearly organized. Tables are extensive and detailed, allowing readers to analyze the data independently. However, each table would benefit from a brief accompanying statement that highlights the key finding. In the "Binge eating symptoms" section, it is worth adding a graphical presentation of the results.
Answer: We have added a brief interpretative statement below each table to emphasize the main finding. In addition, as suggested for the “Binge eating symptoms” section, we have included a graphical representation of the results (Figure 2).
Point 4 – Discussion
The discussion is substantively strong and well-argued. Its weaknesses include length and repetitiveness; it's worth focusing on the most important elements. Practical implications could be further emphasized – for example, whether the results suggest that nutritional interventions should focus on increasing the consumption of unprocessed foods.
Answer: We have changed the Discussion by removing repetitive sections and condensing overlapping arguments to improve clarity and focus. We have also emphasized the main findings more explicitly, ensuring that the discussion centers on the most relevant implications of the results. Additionally, we have expanded the section on practical implications, particularly regarding how nutritional interventions may benefit from prioritizing the promotion of unprocessed and minimally processed foods among adolescents with obesity.
Point 5 – Conclusions Overall, the conclusions are supported by the results but do not fully reflect all findings. It should be clearly stated that the results apply only to this specific group. A stronger link between the conclusions and the methodological limitations would enhance their credibility.
Answer: We revised the conclusion section to explicitly state that the results apply to adolescents with obesity enrolled in this specific clinical program, and should not be generalized to broader populations. In addition, we strengthened the connection between the conclusions and the methodological limitations, particularly regarding the sample size, study design, and restricted external validity. This revision aims to improve the transparency and credibility of the final statements.
Point 6- The tables are factually correct and complete, a graphical presentation of the results would improve the quality of the work. The literature is correct, consisting mainly of newer reports. It is worth supplementing the abstract with the date of the research.
Answer: We thank the reviewer for the constructive feedback. We agree that a graphical representation would complement the tables and improve the readability of the results. Therefore, we have added a figure summarizing the key findings (Graphical abstract) to provide a visual overview of the main associations observed. As suggested, we have also updated the abstract to include the period during which the data were collected.
Round 2
Reviewer 1 Report
Comments and Suggestions for Authors